# Re-Analysis of the Widely Used Recombinant Murine Cytomegalovirus MCMV-m157luc Derived from the Bacmid pSM3fr Confirms Its Hybrid Nature

**DOI:** 10.3390/ijms241814102

**Published:** 2023-09-14

**Authors:** Arne Cordsmeier, Christopher Bednar, Sabrina Kübel, Larissa Bauer, Armin Ensser

**Affiliations:** Institute for Clinical and Molecular Virology, Friedrich-Alexander-Universität Erlangen-Nürnberg, 91054 Erlangen, Germany; arne.cordsmeier@uk-erlangen.de (A.C.); christopher.bednar@uk-erlangen.de (C.B.); sabrina.kuebel@uk-erlangen.de (S.K.); larissa.bauer@uk-erlangen.de (L.B.)

**Keywords:** MCMV, pSM3fr, MCMV-m157lucMCK-2 repair, MCMV bacmid reference

## Abstract

Murine cytomegalovirus (MCMV), and, in particular, recombinant virus derived from MCMV-bacmid pSM3fr, is widely used as the small animal infection model for human cytomegalovirus (HCMV). We sequenced the complete genomes of MCMV strains and recombinants for quality control. However, we noticed deviances from the deposited reference sequences of MCMV-bacmid pSM3fr. This prompted us to re-analyze pSM3fr and reannotate the reference sequence, as well as that for the commonly used MCMV-m157luc reporter virus. A correct reference sequence for this frequently used pSM3fr, containing a repaired version of m129 (MCK-2) and the luciferase gene instead of ORF m157, was constructed. The new reference also contains the original bacmid sequence, and it has a hybrid origin from MCMV strains Smith and K181.

## 1. Introduction

Human cytomegalovirus (HCMV) is a widely distributed pathogen, with seroprevalence of the lifelong infection varying from 40% to more than 90%. HCMV is the most frequent congenital pathogen after infection *in utero* and a significant clinical problem in immunocompromised hosts, causing severe symptoms like pneumonia, retinitis, and hepatitis [1]. Research on HCMV is complicated by the strict tropism of the virus for human cells and tissue, which makes it necessary to use surrogates like related betaherpesviruses from model animals [2,3,4], or more recently, complicated xenograft models like those in immune-deficient mice [5]. Murine cytomegalovirus (MCMV) has a high sequence similarity to HCMV and, not least due to availability of transgenic hosts, is widely used in small animals as an infection model for HCMV. The route of infection of MCMV was analyzed by utilizing reporter viruses [6,7], showing that MCMV spreads from the initial nasal entrance site to secondary vascular sites (e.g., spleen) and subsequently to tertiary sites like the salivary glands. Furthermore, it was demonstrated that the primary target after intra-footpad infection is the popliteal lymph node, which is neither host- nor virus-strain-specific. Intra-footpad infection can thus simulate peripheral infection [6]. Experiments analyzing the natural oral and nasal route of infection show that, in this case, the olfactory neurons are the primary target cells. However, deeper infection of the lower respiratory tract is also possible. After nasal infection, the tertiary virus spreads to the salivary glands and therein establishes persistence [7]. The replication of CMV usually occurs in differentiated myeloid cells [8]. Typical laboratory strains of MCMV are the strains Smith and K181. The genomes of these strains show a very high sequence similarity to each other (nucleic acid identity > 99%). However, they are highly variable with respect to “private” gene families like *m*02 and *m*145, located near the opposite termini of the genome. These play an important role in viral antigen presentation and, therefore, immune evasion ([9], see also Figure 1). The MCMV Smith and K181 strains differ in their infection dynamics *in vitro* and *in vivo*. *In vitro*, K181 leads to lower viral titers and smaller plaques in comparison to Smith, while the titers of K181 *in vivo* are higher in salivary glands [10,11]. In general, there are distinct genetic differences in the CMV genomes infecting different hosts, and also between MCMV strains, which are often found in the area of surface proteins, suggesting host adaption and immune evasion [12,13]. 

Genetic modification of viral genomes with traditional methods is difficult and laborious. The first bacterial artificial chromosome (bacmid) containing a near-complete sequence of a herpesvirus, MCMV strain Smith by Messerle and colleagues [14], represented a major breakthrough. However, extensive analysis of this bacmid pSM3 detected a gap comprising the open reading frames (ORFs) m151-m158. The loss of this region might have occurred to reduce the oversized nature of the viral genome due to insertion of the bacmid sequence via recombination. The missing region was later reinserted using homologous recombination with cloned MCMV DNA sequences [15] derived from plasmids initially designated as strain Smith but later identified as being derived from MCMV strain K181 [16,17]. Therefore, the resulting bacmid pSM3fr represents a hybrid consisting of the *m*150 to *m*158 genes belonging to the K181 strain within the genome of strain Smith. Thereafter, several groups observed an impaired replication of the pSM3fr-derived viruses in salivary glands; this was finally shown to be attributable to a frameshift mutation in the MCK-2 (MCMV chemokine homologue) gene, encoded in the ORF m129, leading to impaired replication in salivary glands. Repair of the *m*129 frameshift in pSM3fr-MCK-2fl (full length) restored salivary gland replication [18]. Interestingly, a further gene, m155, was initially implicated in the deficiency in salivary gland replication of the pSM3fr-derived viruses [19]. Furthermore, the pSM3fr bacmid was used as a template for generation of various reporter viruses, e.g., MCMV-m157luc, with luciferase instead of m157. The M157 glycoprotein was identified as a ligand for the natural killer (NK) cell activation receptor Ly49H [20]. Interestingly, this interaction between a viral protein and a host cell receptor was shown to be the opposite of immune evasion in the scope of co-infection and mutation studies [21,22]. Therefore, the deletion of the *m*157 gene with subsequent replacement by a luciferase gene does not impair the fitness of the resulting viruses during infection but sensitizes C57BL6 mice due to reduced NK cell defense. In other mouse strains, which are Ly49H-negative, M157 might act as ligand for different receptors. The original pSM3fr bacmid and derivatives like MCMV-m157luc were used for multiple studies (e.g., [23,24]); MCMV-m157luc was also independently repaired in the *m*129 gene encoding MCK-2 [25]. In the scope of another project from our lab, we sequenced the full bacmids and viruses derived thereof, and we noted numerous divergences from the deposited m129-repaired pSM3fr-MCK-2fl reference sequence in GenBank (Acc. No. KY348373), which prompted us to reassess the history of this widely used reagent. During our effort to reconstruct the history of this important reagent, we generated new reference sequences for the original pSM3fr and the widely used luciferase-containing bacmids.

## 2. Results and Discussion

In the scope of generating modified MCMV bacmids derived from the widely used pSM3fr bacmid, we performed full-genome sequencing of the constructed DNA via next-generation sequencing (NGS) to verify the correct assembly and search for potential mutations. Since we detected a high number of unexpected mismatches in specific regions, we also sequenced the original pSM3fr bacmid [14,17].

The sequences generated from the original pSM3fr bacmid were mapped to the MCMV strain Smith (GenBank Acc. No.: OP429142) and the deposited pSM3fr-MCK-2fl reference sequence (GenBank Acc. No.: KY348373) with average coverages of 160 and 162, respectively. Interestingly, for both mappings, the variant detection showed a heavily mutated region in the open reading frames (ORFs) m150 and m151, which encode type 1 membrane proteins of the *m*145 family (Figure 2A). In total, for m150, we detected 19 SNPs that lead to 12 amino acid changes, and for m151, 51 SNPs that lead to 32 amino acid changes. The alignment of the consensus sequence of m150 and m151 with the same region of different MCMV strains revealed that these ORFs match the strain K181, while the rest of the sequences match the Smith strain (Figure 2B,C). These regions are, therefore, part of the reinserted ORFs to complete the MCMV genome within the bacmid, in order to repair an initially loss of m151-m158 during cloning [17]. This divergence was expected for the comparison with the Smith strain, but not for the deposited pSM3fr-MCK-2fl reference sequence [18]. Moreover, this reference sequence does not contain annotations for the bacmid sequences, which were found in our mapping subsequent to ORF m158, flanked by repetitive elements for recombination (Figure 3A). Lodha et al. recently studied the transcriptional profile of MCMV, including noncoding RNA of the pSM3fr-derived virus; in this context, they also adapted the deposited reference sequence KY348373 [26]. However, this sequence does not contain the corrections in the region m150 to m158. We constructed a new reference sequence for the hybrid pSM3fr bacmid, which contains the regions originating from K181 at the position of the ORFs m150 to m158, the frameshift mutation in m129 (encoding for MCMV chemokine homologue MCK-2 [18]), as well as the bacmid-specific regions between m158 and m159 (Figure 3B). Further variants that were detected after mapping our pSM3fr libraries to the deposited reference sequence (KY348373) are listed in Table 1. Of note, this earlier deposited pSM3fr MCK2-fl reference sequence was obviously corrected for the *m*129 frameshift, but not for any of the further variant positions with respect to strain Smith described in [18]. 

Another derivative of the widely used bacmid pSM3fr derived from MCMV contains a repaired version of the MCK-2 gene (*m*129) [18] and a HCMV IE promoter-driven luciferase cassette replacing ORF m157 (MCMV-m157luc) [23,24]. For future analysis, we also generated an annotated reference sequence for this bacmid derived from respective consensus sequences and confirmed it via resequencing (Figure 3C). 

Furthermore, we sequenced two different stocks of low-passage viruses reconstituted from the MCMV-m157luc bacmid with an average coverage of 330. The resulting reads were then mapped to our newly generated reference sequence for MCMV-m157luc (Figure 4) and reached average coverages of 256 and 2523, respectively. Interestingly, both mappings show gaps within the area encoding the bacmid components. This indicates that loss of genomic regions unimportant for viral replication is not infrequent, most probably due to the large size of the MCMV genome, which is already at the edge of the nucleocapsid packaging capacity [17,27]. This probably was also the cause of the initially observed deletion of m151 to m158 in the original bacmid [14]. Since the genes encoded in the bacmid areas are dispensable for viral replication and infection, we suggest that viruses that lose these parts—instead of parts of the original viral genome—have replication advantages over viruses with the full bacmid. This is supported by the fact that we observe partial bacmid loss in two different viral stocks at different positions, but no gene loss or extensive gaps in other parts of the genome. Nevertheless, the tendency of gene loss and further genetic alterations during passaging demonstrates the usefulness of regular full-genome sequencing of both, newly generated viral stocks as well as passaged viruses. It also suggests that one should more regularly make use of the two loxP sites flanking the bacmid sequences; by passaging the virus through Cre recombinase-expressing cells, better comparable viruses with a homogenous deletion of the bacmid and a better packable genome size may be obtained. However, the loxP sites are not at the very ends of the bacmid, leaving residual foreign sequences. Alternatively, the vector cassette could be moved to the end of the genome or placed within an essential gene to increase selection pressure [28]. Since HCMV has a similar genome size to MCMV, we would assume similar parameters and restrictions apply for bacmid cloning and rescue. This indicates that particular attention is required in the characterization of recombinant herpesviruses, like marker rescue, careful restriction analysis and resequencing of full recombinant genomes.

Furthermore, since HCMV has a similar genome size to MCMV, we would expect a tendency of gene loss in bacmid systems as well and would recommend full-genome sequencing of bacmid-derived HMCV.

In conclusion, we provide the CMV research community with new and corrected references for the extensively used pSM3fr bacmid (accession: ERZ20801763) and the MCK-2-repaired version, MCMV-m157luc reporter virus, containing a luciferase instead of ORF m157 (accession: ERZ20801746).

## 3. Material and Methods

### 3.1. Bacterial Culture and DNA Preparation

*Escherichia coli* DH10B-derived EL250 [29] or GS1783 [30] carrying bacmid DNA was cultured overnight in LB medium containing 15 µg/mL chloramphenicol at 32 °C and 200 rpm. Bacmid DNA was purified using the PureLink^TM^ HiPure Plasmid Maxiprep Kit of Invitrogen, Thermo Fisher Scientific, Waltham, MA, USA (K210006) according to the manufacturer’s instructions.

### 3.2. Library Preparation and Sequencing

Purified bacmid DNA was obtained via standard precipitation protocols. Viral DNA was extracted utilizing a Qiagen (Venlo, The Netherlands) EZ-1 instrument. For the library preparation for next-generation sequencing, 500 ng of bacmid DNA or extracted viral DNA was fragmented and adapters and indices were ligated using the NEBNext^®^ Ultra™ II FS DNA Library Prep Kit from Illumina, San Diego, CA, USA (E7805), according to manufacturer’s instructions, with a fragmentation time of 15 min.

Sequencing was performed via paired-end sequencing utilizing the MiSeq Reagent Kit v3 (150 cycles) on a MiSeq™ Instrument (Illumina, San Diego, CA, USA). Sequence analysis was conducted using CLC Genomics Workbenches 22 and 23 (Qiagen Aarhus A/S, Denmark). Raw reads were trimmed for quality (limit 0.05, Mott trimming algorithm), adapters, and ambiguities. The trimmed reads were mapped with different stringencies, testing for the best distribution. Duplicate reads were removed from the mapping. Variants from the reference sequence were detected, only considering regions with coverages above 50 and a frequency of the variant in the reads of more than 80%. 

### 3.3. Production and Purification of MCMV Stocks

In order to generate high-titer stocks of MCMV, murine embryonic fibroblast (MEF) cells were seeded 1:2 into twelve T175 cell culture flasks. On the next day, purified MCMV virions were thawed and 1 × 10^5^ virions were added to 25 mL of complete DMEM. After extensive vortexing, the MEF supernatant of each flask was exchanged for 25 mL of complete DMEM containing MCMV virions and the cells were incubated for 1 week under cell culture conditions. Cell culture supernatants were centrifuged (1500× *g*, 20 min, 4 °C) to remove cell debris. The supernatants were ultracentrifuged (33,000× *g*, 3 h, 4 °C). The supernatant was discarded; then, the pellets were resuspended in a few microliters of residual media, pooled, and dispersed using a syringe. Following that, 1 mL of concentrated virions was carefully loaded on a 9 mL cushion of 15% sucrose in virus suspension buffer (VSP, 50 mM Tris-HCl (pH 7.8), 12 mM KCl, 5 mM EDTA) and purified via ultracentrifugation (70,000× *g*, 1 h, 4 °C). The sucrose cushion was discarded; then, the pellets were washed in 10 mL of PBS and repelleted (70,000× *g*, 1 h, 4 °C). PBS was discarded and 500 µL of VSP was added and incubated on ice overnight. The next day, pellets were resuspended, pooled, dispersed using a syringe, and stored at −80 °C in 40 µL aliquots.

### 3.4. Sequences

Reference sequences were deposited in the European Nucleotide Archive (https://www.ebi.ac.uk/ena/browser/home, accessed on 31 July 2023). The accession number for the original pSM3fr bacmid is ERZ20801763, while the accession number for the MCK-2 repaired version with a luciferase instead of ORF m157 is ERZ20801746.

## Figures and Tables

**Figure 1 ijms-24-14102-f001:**
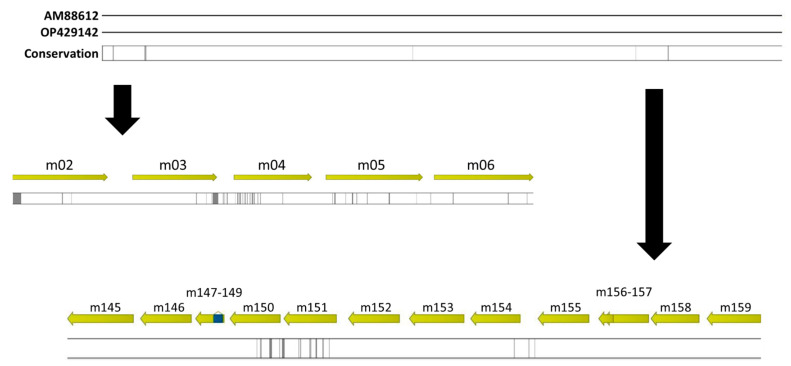
**Genome alignment of the MCMV strains Smith and K181.** The reference sequences for the MCMV strains Smith (OP429142) and K181 (AM88612) from GenBank were aligned and sequence conservation is displayed. Black bars indicate lower conservation. The genome areas of *m*02-*m*06 as well as *m*145-*m*159 are enlarged to demonstrate high sequence diversity in these respective areas between the two strains.

**Figure 2 ijms-24-14102-f002:**
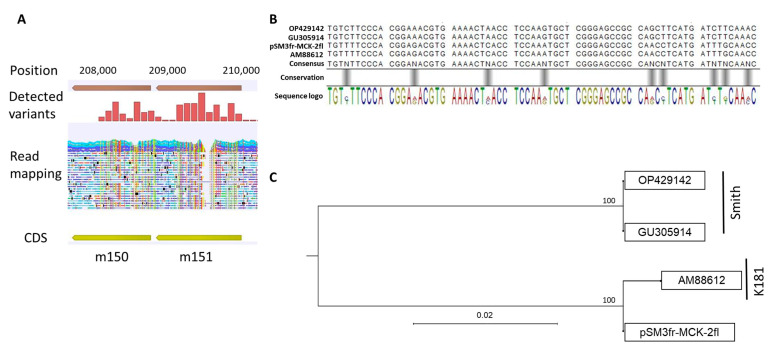
**Comparison of ORFs m150 and m151 between MCMV strains Smith and K181.** (**A**) Mapping of original pSM3fr reads against MCMV strain Smith and pSM3fr-MCK-2fl reference sequence and variant detection. Colored vertical lines indicate multiple nucleotide polymorphisms. (**B**) Section of an alignment of pSM3fr-MCK-2fl sequence of ORFs m151 and m152 as well as the respective regions of database reference sequences for MCMV strains Smith (OP429142, GU305914) and K181 (AM88612). Alignment was constructed using the Jukes–Cantor model. Black bars indicate lower conservation. (**C**) Maximum-likelihood tree generation with 1000 bootstrap repeats from the alignment of (**B**); numbers at bifurcation give percent of trees with the respective branching. Scale bars indicate the phylogenetic distance in percent of number of substitutions/changes per nucleotide.

**Figure 3 ijms-24-14102-f003:**
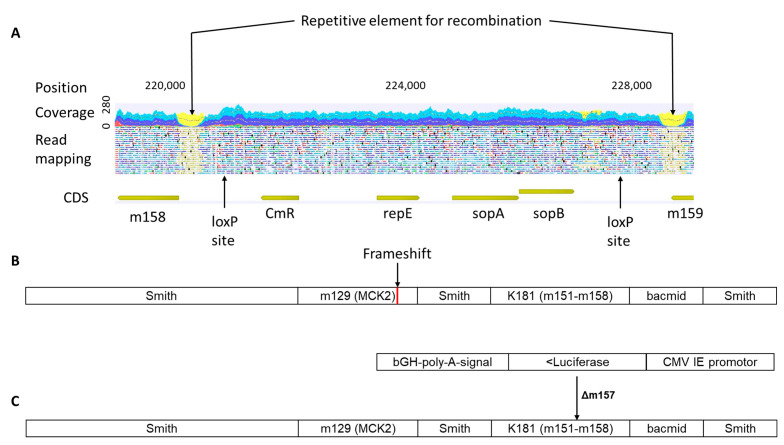
**Schematic representation of corrected reference sequences.** (**A**) Mapping showing the bacmid-specific sequences within the references. (**B**) The corrected pSM3fr reference sequence, with the frameshift mutation in *m*129 and strain-K181-derived sequences in *m*151-*m*158. (**C**) The newly generated reference sequence for the repaired and modified pSM3fr-m157luc bacmid, containing the repaired version of m129 and a luciferase gene cassette instead of m157.

**Figure 4 ijms-24-14102-f004:**
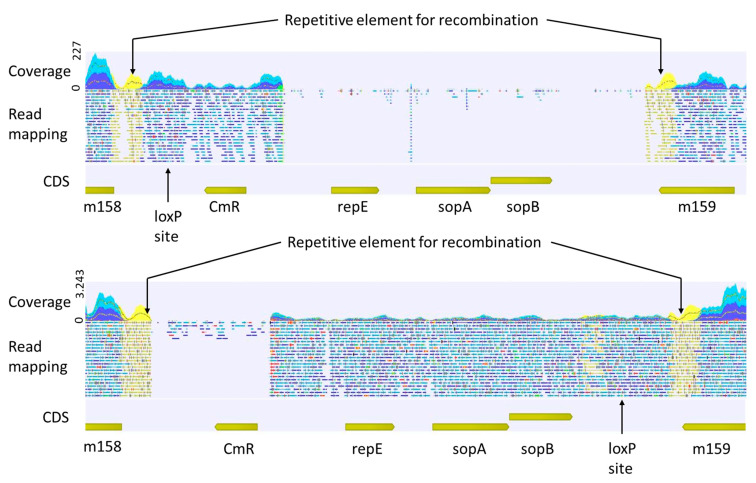
**Mapping of reads obtained from low-passage MCMV viruses (m157-luc).** Displayed are the mapped reads from two different viral stocks. The genome area spanning the bacmid components, which are flanked by the repetitive elements for recombination, is shown. In both cases, gaps (without specific read mapping or with strongly reduced specific read mapping) can be observed.

**Table 1 ijms-24-14102-t001:** **Detected variants of our pSM3fr mapping to the deposited pSM3fr reference sequence (KY348373).** Only variants that are observed in above 80% of the reads are shown. ORF = open reading frame, fs = frameshift, ins = insertion, - = gap, SNV = single-nucleotide variant, MNV = multi-nucleotide variant, ? = unknown effect.

ORF	Position	Type	Frequency [%]	Amino Acid Change	In Reference	In Mapping
**Variants already described in** [18] **but not corrected in the sequence**						
m20	20957	Deletion	97	Q697fs	G	-
intergenic	31973	Insertion	91	none	-	G
m29, m29.1	36197	Insertion	99	S297fs; D155fs	-	G
intergenic	37259	Insertion	84	none	-	C
M31	38802	Insertion	87	A509fs	-	G
m45.1	61917	Insertion	88	A375fs	-	C
m58	92088	Deletion	96	N110fs	A	-
m58	92358	Deletion	100	P200fs	C	-
m142.1	201402	Insertion	93	G477fs	-	G
intergenic	214373	Deletion	99	none	T	-
m156	215809	Insertion	87	Insertion of P between V56 and S57	-	CGG
intergenic	219102	Insertion	90	none	-	A
intergenic	219308	SNV	98	none	C	A
intergenic	219496	SNV	97	none	G	A
intergenic	221004	Insertion	94	none	-	G
intergenic	221024	Insertion	87	none	-	GG
intergenic	226294	Deletion	98	none	C	-
m161	228998	Insertion	85	S187fs	-	A
**Variants due to the hybrid nature of the bacmid (K181 from m150 to m158)**						
m150	208207	SNV	96	K195Q	T	G
m150	208211	SNV	93	none	A	C
m150	208214	SNV	95	none	C	T
m150	208235	SNV	94	none	A	G
m150	208243	MNV	99	Replacement of E182 and V183 by D and I	CC	TA
m150	208250	MNV	98	R180N	ACG	GTT
m150	208261	SNV	98	V177L	C	A
m150	208280	SNV	100	none	G	C
m150	208283	SNV	98	none	G	A
m150	208289	SNV	100	none	T	C
m150	208500	Insertion	99	W97fs	-	G
m150	208502	Deletion	96	W97fs	C	-
m150	208503	Insertion	88	L96fs	-	T
m150	208505	Deletion	85	L96fs	A	-
m150	208508	SNV	88	none	A	G
m150	208510	SNV	93	H94Y	G	A
m150	208525	SNV	98	D89N	C	T
m150	208529	MNV	98	K87T	CT	AG
m150	208788	Insertion	90	M1?	-	T
m151	209113	Insertion	72	E290 duplicate	-	TTC
m151	209126	Deletion	98	A287fs	A	-
m151	209127	Insertion	97	A286fs	-	T
m151	209129	SNV	98	none	A	G
m151	209138	SNV	99	none	C	T
m151	209154	MNV	99	T277V	GT	AC
m151	209157	SNV	100	A276V	G	A
m151	209161	SNV	100	A275T	C	T
m151	209258	SNV	98	none	T	C
m151	209262	SNV	96	S241W	G	C
m151	209273	SNV	99	N237K	A	C
m151	209282	SNV	99	none	A	G
m151	209289	SNV	100	V232A	A	G
m151	209291	SNV	98	none	A	C
m151	209294	SNV	98	none	T	C
m151	209348	SNV	93	none	T	G
m151	209354	SNV	96	none	G	A
m151	209357	SNV	99	none	T	A
m151	209359	MNV	97	Replacement of A208 and T209 by V and A	TTG	CCA
m151	209375	SNV	90	none	T	C
m151	209379	MNV	97	N202G	TT	CC
m151	209383	SNV	99	T201S	T	A
m151	209399	MNV	99	S195F	GG	AA
m151	209403	MNV	97	V194T	ACC	GTT
m151	209406	Insertion	97	S193fs	-	T
m151	209408	Deletion	94	S193fs	C	-
m151	209414	SNV	96	none	T	C
m151	209416	SNV	99	K190E	T	C
m151	209419	SNV	94	T189A	T	C
m151	209437	MNV	100	L183I	GC	TA
m151	209440	SNV	96	T182S	T	A
m151	209446	MNV	98	I180L	TG	AA
m151	209452	SNV	100	Q178K	G	T
m151	209457	SNV	94	R176K	C	T
m151	209459	SNV	93	none	G	A
m151	209461	Deletion	97	R175fs	G	-
m151	209462	Insertion	100	R175fs	-	G
m151	209467	MNV	100	I173L	TA	GG
m151	209471	SNV	100	none	A	T
m151	209474	SNV	100	none	A	G
m151	209576	Insertion	73	P138fs	-	AC
m151	209579	Replacement	92	I135fs	TA	C
m151	209583	Replacement	100	L134fs	AA	G
m151	209588	MNV	98	A132E	GG	CT
m151	209597	SNV	93	none	G	C
m151	209603	SNV	99	none	A	G
m151	209605	Insertion	94	Y127fs	-	G
m151	209607	Deletion	97	Y126fs	T	-
m151	209801	SNV	96	none	G	C
m151	209815	SNV	97	D57N	C	T
m151	209818	MNV	98	Y56H	AC	GT

## Data Availability

The generated reference sequences were deposited in the European Nucleotide Archive (https://www.ebi.ac.uk/ena/browser/home, accessed on 31 July 2023). The accession number for the original pSM3fr bacmid is ERZ20801763, while the accession number for the MCK-2 repaired version with a luciferase instead of ORF m157 is ERZ20801746. All other data are available upon request from the corresponding author.

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
