# Peer review of "Re-Analysis of the Widely Used Recombinant Murine Cytomegalovirus MCMV-m157luc Derived from the Bacmid pSM3fr Confirms Its Hybrid Nature"

_ijms, 2023, doi:10.3390/ijms241814102_

Round 1

Reviewer 1 Report

Line 24,  39 and 44: no need to write "reviewed in"

table 1 - should be in the supplementary material 

Author Response

We thank the Reviewer 1 who found that we present the data appropriate and conclusive.

Regarding the specific points/comments/suggestions

-Line 24,  39 and 44: no need to write "reviewed in"

 We have removed these prefixes

-table 1 - should be in the supplementary material

The table has not been moved, since supplementary material is not intended for a brief report

Reviewer 2 Report

Mouse CMV is an important model for analyzing various aspects of CMV pathogenesis. Cordsmeier et al. have sequenced the genomes (and bacmids) of the recombinant MCMV-m157luc as well as the parental bacmid pSM3fr and generated a new reference sequence deposited in the EBI databank. This is highly appreciated because (i) these are widely used (model) viruses and (ii) the previously available Genbank entry for pSM3fr (and other bacmids) apparently carried a number of errors. The sequence analysis and the representation of the results were well done and I have only a few minor suggestions.

Specific points:

- Fig. 2. Although given in table 1, it would be of interest to depict how the nucleotide diversity between the Smith and K181 strains translates to amino acid changes in m150 and m151 (and tell in the text).

- Fig. 3A. There is no labelling of the CDS to the very left (applies to Fig. 4 as well).

- Fig. 4 and text, lines 118-134. It is an important observation that the excision of the BAC vector sequences by homologous recombination is not completely efficient (perhaps predictable in view of the data shown in ref. 15). Nevertheless, it is a concern with respect to the generation of homogeneous preparations of recombinant viruses and is of interest to the CMV community.

In line 122, ref. 15 may be added to ref. 24 because the argument for selection against over-length genomes was probably already mentioned in this publication too. loxP-mediated excision of the bacmid vector sequences would be an alternative as indicated by the authors, and has in fact be used for other herpesviruses. However, for the pSM3fr bacmid it would probably require additional modifications as it seems that the loxP sites are currently not located at the very ends of the bacmid vector, possibly leaving some foreign sequences or even disrupting open reading frames. Other options would be to move the vector cassette to the very end of the genome (between the pac sites) or within essential genes (to increase the selection pressure for excision) as done for HCMV (by Tischer and Sinzger), or perhaps to increase the chance of homologous recombination by inducing a double strand break into the transfected bacmids (next to the vector sequences) by various techniques. The authors could elaborate a little bit more on such options.

Minor points:

line 5. The "and" is somewhat displaced.

- lines 51-52 and 122-123. In the original MCMV bacmid pSM3 ORFs m151 to m158 may have been deleted to avoid overlength of the viral genome due to insertion of the bacmid vector insertion.

- lines 65-70. The M157 glycoprotein may act “opposite of immune evasion” in a (small) subset of mouse strains only (such as BL6 mice as indicated). In other mouse strains M157 may act as ligand of inhibitory NK cell receptors. This could be explained in more detail.

- references should be re-checked with respect to style and completeness (e.g., page numbers), particularly since the journal may not offer extensive editing service. As an example, 3 different styles were used for J.Virol., namely J. VIROL. (ref. 15), J Virol (ref. 16), Journal of Virology (ref. 17).

Author Response

We thank the reviewer #2 for the detailed comments and for the appreciation of our efforts and considering the data as important for all who are working with the respective bacmid.

Regarding the specific points/comments/suggestions of the reviewer:

- Fig. 2. Although given in table 1, it would be of interest to depict how the nucleotide diversity between the Smith and K181 strains translates to amino acid changes in m150 and m151 (and tell in the text).

We have added a sentence with information about the number of SNPs and resulting amino acid changes. Lines 95-97

- Fig. 3A. There is no labelling of the CDS to the very left (applies to Fig. 4 as well).

The missing labeling has been added in both figures.

- Fig. 4 and text, lines 118-134. It is an important observation that the excision of the BAC vector sequences by homologous recombination is not completely efficient (perhaps predictable in view of the data shown in ref. 15). Nevertheless, it is a concern with respect to the generation of homogeneous preparations of recombinant viruses and is of interest to the CMV community.

The respective part has been refined.

-In line 122, ref. 15 may be added to ref. 24 because the argument for selection against over-length genomes was probably already mentioned in this publication too. loxP-mediated excision of the bacmid vector sequences would be an alternative as indicated by the authors, and has in fact be used for other herpesviruses. However, for the pSM3fr bacmid it would probably require additional modifications as it seems that the loxP sites are currently not located at the very ends of the bacmid vector, possibly leaving some foreign sequences or even disrupting open reading frames. Other options would be to move the vector cassette to the very end of the genome (between the pac sites) or within essential genes (to increase the selection pressure for excision) as done for HCMV (by Tischer and Sinzger), or perhaps to increase the chance of homologous recombination by inducing a double strand break into the transfected bacmids (next to the vector sequences) by various techniques. The authors could elaborate a little bit more on such options.

Reference 15 has been added. We have also commented on the alternative methods as suggested, citing the example of HCMV. Lines 138-140.

-line 5. The "and" is somewhat displaced.

The mistake has been corrected.

-lines 51-52 and 122-123. In the original MCMV bacmid pSM3 ORFs m151 to m158 may have been deleted to avoid overlength of the viral genome due to insertion of the bacmid vector insertion.

We have added a comment on that in the introduction. Lines 52-54.

-lines 65-70. The M157 glycoprotein may act “opposite of immune evasion” in a (small) subset of mouse strains only (such as BL6 mice as indicated). In other mouse strains M157 may act as ligand of inhibitory NK cell receptors. This could be explained in more detail.

We have added a sentence about LyH49 negative mice. Lines 72-73

-references should be re-checked with respect to style and completeness (e.g., page numbers), particularly since the journal may not offer extensive editing service. As an example, 3 different styles were used for J.Virol., namely J. VIROL

The references have been adapted.

Reviewer 3 Report

An excellent piece of work.

My only suggestions for improvement are as follows:

1. Have any studies been conducted regarding the production of noncoding RNAs in response to infection by the various MCMV strains. As you will know ncRNAs especially miRNAs are implicated in the expression of disease and differential responses my begin to shed some light on this topic.

2. Are there any studies on differences in the surface coat on the MCMV variants?

3. Any thoughts on how your study may apply in the wider context of human CMVs.

The above are merely suggestions to give your study wider interest.

Author Response

We thank the reviewer #3 who found our manuscript an excellent piece of work.

Regarding the comments/suggestions of the reviewer:

-1. Have any studies been conducted regarding the production of noncoding RNAs in response to infection by the various MCMV strains. As you will know ncRNAs especially miRNAs are implicated in the expression of disease and differential responses my begin to shed some light on this topic.

We are not aware of a study comparing ncRNAs of MCMV strains. This is an interesting suggestion. However, generation and addition of these data would exceed the scope of this brief report.

-2. Are there any studies on differences in the surface coat on the MCMV variants?

We are not aware of such studies but this is an interesting issue, also with respect to strain differences.

-3. Any thoughts on how your study may apply in the wider context of human CMVs.

Since HCMV has a similar genome size as MCMV, we would assume similar parameters and restrictions for bacmid cloning and rescue. We have added a comment. Lines 141ff.

Reviewer 4 Report

Review of Manuscript “Re-analysis of the widely used recombinant murine cytomegalovirus MCMV-m157luc derived from the bacmid pSM3fr confirms its hybrid nature”.  

In their brief report, Cordsmeier et al. describe the re-analysis of the murine cytomegalovirus (MCMV) derived bacimid pSM3fr, which in the past has been widely used for small animal models for human cytomegalovirus infection, by next generation sequencing (NGS). In addition to confirming the hybrid nature of the bacimid with sequences originating from both Smith and K181 MCMV strains, they noticed a number of deviances from the published reference sequence mainly localized in the regions of the ORFs m150 and m151 derived from the K181 strain and the bacimid sequences subsequent to m158.

The re-annotations of the pSM3fr bacimid are based on a very solid coverage in the NGS approach as are those for the pSM3fr-m157luc reporter bacimid conformed both by sequencing of the original bacimid and two different stocks from low passage viruses generated from the bacimid.

The results of the re-analysis are presented in a clear form and are thus easy to understand. They are of interest for all researchers working with these bacimids to study the course of CMV infection in small animal models.

English language is fine, only very minor editing required.

Author Response

We thank the reviewer #4 who found our study clearly presented and easy to understand.